# Spatial Transcriptomic Analysis of Surgical Resection Specimens of Primary Head and Neck Squamous Cell Carcinoma Treated with Afatinib in a Window-of-Opportunity Study (EORTC90111-24111)

**DOI:** 10.3390/ijms26051830

**Published:** 2025-02-20

**Authors:** Simon Beyaert, Axelle Loriot, Jean-Pascal Machiels, Sandra Schmitz

**Affiliations:** 1Institut de Recherche Expérimentale et Clinique (IREC), Pôle MIRO, Université Catholique de Louvain (UCLouvain), 1200 Brussels, Belgium; simon.beyaert@saintluc.uclouvain.be (S.B.); jean-pascal.machiels@saintluc.uclouvain.be (J.-P.M.); 2Department of Head & Neck Surgery, Institut Roi Albert II, Cliniques Universitaires Saint-Luc, 1200 Brussels, Belgium; 3Group of Computational Biology and Bioinformatics, de Duve Institute, Université Catholique de Louvain, 1200 Brussels, Belgium; axelle.loriot@uclouvain.be; 4Department of Medical Oncology, Institut Roi Albert II, Cliniques Universitaires Saint-Luc, 1200 Brussels, Belgium

**Keywords:** head and neck squamous cell carcinoma, afatinib, window-of-opportunity study, tumor microenvironment, spatial transcriptomics

## Abstract

Afatinib-induced tumor and microenvironment modifications in head and neck squamous cell carcinoma were evaluated by spatial transcriptomics in surgical specimens and RNA-sequencing in tumor biopsies of patients included in the EORTC-90111-24111 window-of-opportunity study. The aim was to explore tumor evolution and composition under anti-HER therapy. Based on our previous investigations by RNA-seq on tumor biopsies, surgical slides of ID08 and ID15 from the epithelial-to-mesenchymal (EMT) cluster and ID30 from the non-EMT cluster were investigated with spatial transcriptomics. Dimension reduction in ID30 revealed 14 clusters, with clusters overlapping three tumor nodules and the stroma. Differential expression analysis between tumor nodules showed enrichment of the hallmark EMT genelist, with 123 genes in common between the analyses. These genes were involved in PDGF and MET signaling pathways. By comparing gene expression in paired tumor biopsies and the 123 genes from differential analyses obtained in ID30, a list of 13 genes involved in cancer pathways and EMT emerged, which were also highly expressed in ID08 and ID15. These results show a progressive apparition of genes implicated in EMT, MET, and PDGF pathways in tumors after afatinib. Notably, a list of 13 genes emerged which may contain targets to prevent tumor evolution after anti-HER therapy.

## 1. Introduction

With 650,000 cases worldwide, squamous cell carcinoma of the head and neck (HNSCC) is the sixth most common cancer [1]. Curative treatment for early disease (American joint commission on cancer (AJCC) stages I and II) includes radiotherapy or surgery. Standard treatment for stage III and IV disease is multimodal and includes primary chemo-radiation or surgery followed by post-operative (chemo)radiotherapy. In Europe, cetuximab plus platin-based chemotherapy (EXTREME regimen) is administered as first-line treatment for programmed death-ligand (PD-L)1 negative recurrent and/or metastatic (R/M) HNSCC [2]. Cetuximab is a monoclonal antibody targeting the epidermal growth factor receptor (EGFR), which is a member of the human epidermal growth factor (HER)/ErbB family of receptor tyrosine kinases, wherein activation is known to induce survival and tumor growth of cancer cells [3]. However, the benefit of the EXTREME regimen is low, median survival being only 10.1 months [2] with few patients benefiting of the long-term therapeutic effects of these treatments. A better understanding of the resistance mechanisms involved could be beneficial to improve treatment response. Notably, the activation or upregulation of other members of the ErbB receptor family (e.g., following cetuximab [4]) may lead to resistance to anti-EGFR therapy. Because HER2 signaling uses many of the same EGFR downstream effectors, it has been shown to be linked to cetuximab resistance [5]. The fourth member of the family, HER-4, is a transmembrane tyrosine kinase, which can also activate downstream pathways linked to tumor growth. Blocking several HER receptors may, therefore, be of interest.

Afatinib is an oral, irreversible inhibitor of the EGFR, HER2, and HER4 kinases (“pan-HER” inhibitor) and blocks the signaling of all homodimers and heterodimers of the HER family [6]. Afatinib has shown activity in pre-clinical HNSCC studies [7,8] and improves progression-free survival (median = 2.6 months) compared with methotrexate for R/M HNSCC after platinum therapy [9]. We performed a window-of-opportunity (WOO) study with afatinib (EORTC-90111-24111) [10] in primary HNSCC and recently reported our translational findings [11]. In this study, we have shown on paired biopsies that some patients developed epithelial-to-mesenchymal transition (EMT) and activation of cancer-associated fibroblasts (CAFs) after only two weeks of preoperative treatments, while other patients did not. Safety, clinical, and translational data for cetuximab studied in a similar study design (CHIRON study, NCT00714649) showed similar conclusions [12,13]. Emerging techniques for genome-wide spatial transcriptomics hold great promise for producing precise molecular maps that get through bulk RNA-sequencing’s loss of spatial information. Importantly, these technologies utilize an on-slide cDNA synthesis method that records gene expression in the structure of intact tissue, meaning that gene expression from tumor and stroma can be analyzed separately [14]. We analyzed gene expression in three surgical specimens of patients included in the EORTC 90111-24111 WOO study to increase our understanding of gene expression in the context of the spatial organization of afatinib-treated HNSCC.

The goal of the present article is to decipher the molecular behavior of tumor cells and their stroma separately following in vivo exposure to afatinib. Therefore, we explore further spatial transcriptomic data of selected samples of the EORTC-90111-24111 study. The second objective is to confirm data from previous investigations using multiplex immunohistochemistry and bulk RNA-sequencing on tumor biopsies [11], by studying gene expression in the stromal and tumor compartments separately, with a focus on EMT.

## 2. Results

### 2.1. Patients

After careful selection (RNA quality and H&E staining), surgical sections from four patients were selected for spatial transcriptomic analyses. Three patients from the afatinib arm were used, notably two patients (ID08 and ID15) from the cluster 1 (i.e., patients developing EMT following afatinib, as previously described in translational analyses of their paired tumor biopsies [11]) and one patient (ID30) from the cluster 2 (i.e., a patient not developing EMT in post-treatment biopsies [11]). A control patient had been included in the experiments (patient ID14 from study EORTC90111), but after verification of the tissue analyzed on the Visum slide, a procedural error in the experiment rendered the tissue analysis impossible. We, therefore, decided to exclude this patient from the spatial transcriptomic translational analyses. Patient characteristics are available in Table 1

### 2.2. Quality Control

Mean DV200 for the RNA of the three samples included in the analysis was 46.9 (DV200 for ID08, ID15 and ID30 was 53.9, 41.7 and 45.0, respectively). The arbitrary stringent cut-off of 500 expressed genes per spot and 500 UMIs per spot showed that 96.1% of spots from the selected region of interest could be included in the analyses (Figure 1). Unfortunately, in the case of this study, quality control (tumor evaluation on HE staining or DV200) on available FFPE surgical sections from the patients included in the CHIRON study conducted by Schmitz et al. [12] did not allow the realization of spatial transcriptomic analysis.

### 2.3. Dimension Reduction and Clustering

Dimension reduction and clustering of the spots were realized using t-distributed stochastic neighbor embedding (t-SNE) in our samples. Patient ID30, classified in the non-EMT cluster 2 after afatinib in our previous results [11], showed 14 different clusters inside the region of interest (Figure 2A,B). In the ID30 sample, the clustering perfectly matched the tumor compartments (tumor nodules and stroma) visible on the HE section.

Based on the dimension reduction and clustering, we decided to merge clusters in the “tumor nodule (TN) 1” area (top left tumor nodule with clusters 2, 4 and 12); “TN2” (middle tumor nodule with cluster 1 only); “TN3” (bottom right tumor nodule with cluster 9 and 0). Similarly, the TME, or regions surrounding the “tumor nodules” were classified as followed: TME1 (clusters 8, 11, and 13, around TN1) and TME3 (clusters 3 and 7, around TN3). Using the epithelial and fibroblast gene lists of Puram et al. [15], we were able to confirm that the clusters with a high epithelial score were in the subgroups TN 1, 2, and 3 (Figure 2C) and the clusters with a high fibroblast score were in the subgroups TME 1 and 3 (Figure 2D). Cluster 5, 6, and 10 were not included in a subgroup because they were not limited to a single tumor nodule.

In patient ID08 and ID15, classified in the EMT cluster 1 after afatinib in our previous results [11], dimension reduction and clusterization highlighted 13 and 14 clusters, respectively (Appendix A). However, the clusters in these two samples did not overlap precisely the tumor and stromal compartments, probably related to the mixed and heterogeneous morphology of the tissue compartments in tumor undergoing EMT [16].

### 2.4. Differential Expression and Enrichment Analyses in Patient ID30

TN2 vs. TN1 differential expression analyses showed 394 differentially expressed genes, including 313 upregulated genes (e.g., *COL1A1*, *COL1A2*, *POSTN*, *PDGFRA*, *FN1*, *SNAI2*, *SPARC*, …) and 81 downregulated genes (e.g., *KRT13*, *KRT6B*, *SPRR3*, *SPRR2A*, …) in TN2 compared to TN1 (Figure 3A,B and Appendix A). ORA of upregulated genes showed significant activation of the hallmark epithelial–mesenchymal transition (*p* < 0.001) and the hallmark myogenesis (*p* < 0.05) gene lists. ORA of downregulated genes showed significant activation of the hallmark myc targets V1 (*p* < 0.05) gene list.

TN3 vs. TN1 differential expression analyses showed 765 differentially expressed genes, including 476 upregulated genes (e.g., *COL1A1*, *COL1A2*, *FN1*, *PDGFRA*, *POSTN*, *SPARC*, *CXCL9*, …) and 289 downregulated genes (e.g., *IGKC*, *KRT13*, *S100A8*, *SPRR3*) in TN3 compared to TN1 (Figure 3C,D and Appendix A). ORA of upregulated genes showed significant activation of the hallmark epithelial–mesenchymal transition (*p* < 0.05) and the hallmark hypoxia (*p* < 0.05) gene lists. No gene list emerged from the enrichment analysis of downregulated genes.

Finally, 123 genes were commonly upregulated in the differential expression analysis described above (TN2 vs. TN1 and TN3 vs. TN1) (Appendix A and Appendix A). The g:Profiler enrichment analysis showed that these genes were involved, notably, in extracellular matrix component, cell adhesion, platelet-derived growth factor (PDGF) binding and MET pathway (Appendix A). In addition, these EMT-related genes highly expressed in the differential expression analyses described in patient ID30 are preferentially expressed in contact with normal tissue (depth) in tumors ID08 and ID15 (Figure 4). Furthermore, by comparing the upregulated genes (i.e., *p* < 0.05 and log2fold change > 0) from bulk RNAseq analysis in post vs. pre-afatinib tumor biopsies and the tumor nodules differential expression analyses, 13 genes in common were discovered: *COL1A1*, *COL1A2*, *POSTN*, *PDGFRA*, *SLC38A11*, *SPARC*, *VWA5A*, *MUC4*, *SLC9A9*, *CXCL9*, *DCLK1*, *ZNF652* and *ASPN* (Appendix A and Appendix A). This gene list is highly expressed in TN2 and TN3 of patient ID30 and in the depth, close to the invasion margin, of tumors from patients ID08 and ID15 as well (Figure 5). This list of 13 genes is also upregulated after afatinib in tumor biopsies from the 13 available patients as described in our previous translational findings (Appendix A).

Similarly, TME3 vs. TME1 differential expression analyses showed 900 differentially expressed genes, including 858 upregulated genes (e.g., *MMP11*, *POSTN*, *FN1*, *COL1A1*, *CXCL9*, *CXCL14*) and 42 downregulated genes (e.g., *KRT13*, *S100A8*, *S100A9*) in TME3 compared to TME1 (Figure 6A–D and Appendix A). ORA of upregulated genes showed significant activation of the hallmark epithelial–mesenchymal transition (*p* < 0.05). On the other hand, ORA of the downregulated genes showed significant activation of the hallmark epithelial–mesenchymal transition (*p* < 0.001), hallmark hypoxia (*p* < 0.05), hallmark coagulation (*p* < 0.05), hallmark apoptosis (*p* < 0.05) and hallmark estrogen response late (*p* < 0.05) gene lists.

No differential gene expression analysis was performed for ID08 and ID15. But as previously described [11], an HNSCC-adapted mesenchymal and epithelial gene list [17] was highly and weakly expressed in these tumors, respectively.

## 3. Discussion

In line with our previous findings [11], EMT-related genes were identified to be highly expressed in surgical sections of patient ID08 and ID15, analyzed by spatial transcriptomics, particularly in the invasive margin. However, patient ID30, initially clustered as a patient not developing EMT after afatinib based on bulk RNA-seq analysis on paired tumor biopsy analyses [11], also showed EMT-related genes occurring in TN2 and TN3 compared to TN1 in the surgical specimen. This is particularly interesting and a demonstration of tumor heterogeneity leading to progressive intra-tumoral and intra-individual modifications after exposition to a therapeutic agent. For this reason, performing analysis on the surgical specimen is clearly an advantage when compared to analysis on a single biopsy. Spatial transcriptomics has enabled us to study the whole tumor and to identify that EMT and non EMT patterns can be present inside a same tumor as for ID30. The presence of other TNs expressing EMT genes showed that patient ID30’s tumor could also potentially have changed into a more mixed and heterogeneous tissue composed of oligoclusters of mesenchymal-like cancer cells, like in patients ID08 and ID15. The patients included in the EORTC90111-24111 study had only two weeks of treatment by afatinib in order to avoid unethical delay in standard curative surgery [10]. Prolonged pre-operative afatinib could have induced EMT in the whole tumor of patient ID30. EMT is a dynamic process that has several transition states [18], and process times between states may vary from tumor to tumor. Indeed, within the EMT transcription factors [18], only *SNAI2* was overexpressed in TN2 compared to TN1 in patient ID30, which is known to reach its peak expression relatively early in the EMT process [15,18].

Our analyses show the presence of several therapeutic targets that can promote the onset of EMT transition. Notably, 123 genes were commonly upregulated in the differential expression analysis described between TN2 vs. TN1 and TN3 vs. TN1. Enrichment analyses of these 123 upregulated genes show the activation of MET and PDGF-related signaling pathways. As already suggested in our previous results [11], targeting multiple tyrosine kinase receptors may overcome resistance to tyrosine kinase inhibitors and may be of interest [19]. Our previous results [11] showed hyperexpression of hepatocyte growth factor (*HGF*), MET’s main ligand [20], and *PDGFRA/PDGFRB* in post-afatinib biopsies in patients classified in the EMT cluster, highlighting the interest of these targets in afatinib-induced EMT. Indeed, Yi et al. [21] have shown that inhibition of HGF/cMet signaling in lung cancer cell lines prevented CAF-induced EMT and EGFR-tyrosine kinase inhibitor resistance. Furthermore, it has been found that tumor cell lines with a high EMT score are, interestingly, responsive to PDGFR inhibitors yet resistant to EGFR inhibitors [22]. Indeed, recent data from the literature of a phase 2 trial [23] investigated the efficacy of ficlatuzumab (monoclonal anti-HGF antibody) in combination with cetuximab in patients with pan-refractory R/M HNSCC. Patients (n = 33) showed progression-free survival of 3.7 months with an objective response rate of 19%, including two complete and four partial responses. Better survival was shown in patients with cMet overexpression. To the best of our knowledge, the efficacy of a combination of anti-HER drug and anti-PDGF(R) therapies has, however, not yet been investigated [24]. More pre-clinical data on the efficacy of these anti-cancer treatment combinations could be of great interest before initiating phase 1 safety studies in patients. Among the 123 genes described above, 13 were in common with differential expression analyses of tumor nodules in ID30 and post- vs. pre-afatinib tumor biopsies [11]. Theses 13 genes (e.g., *COL1A1*, *POSTN*, *ASPN*, *PDGFRA*, *MUC4*, *SPARC*), implicated notably in EMT and cancer progression [25,26,27,28,29,30,31], were found to be highly expressed in depth of tumor ID15 and ID08, near the invasive margin. Interestingly, in a similar window-of-opportunity study where patients received cetuximab before surgery, several common genes were also upregulated after cetuximab, such as *ASPN*, *MUC4*, and PDGF-related genes [13]. Therefore, targets of choice may be included in this gene list.

Periostin, coded by *POSTN*, is a matricellular protein, which can be found in CAFs or epithelial cancer cells in HNSCC [32,33]. The binding of periostin to integrins αvβ3 and αvβ5 on malignant cells triggers the FAK, Phosphoinositide 3-kinase (PI3K), and Akt signaling pathways, resulting in cell migration [32]. In vitro experiments showed that *POSTN* was expressed in the HNSCC cell line with high partial-EMT score [34]. In addition, periostin overexpression in HNSCC cells induced invasion both in vitro and in vivo [35]. In solid tumors, high *POSTN* expression is associated with a more aggressive tumor behavior, advanced stage, and poor prognosis [29]. In a breast cancer mouse model, anti-periostin antibody inhibited primary tumor growth, metastatic lesions, and increased the survival rate [36], making periostin a potential target, notably for blocking the partial-EMT state. Secondly, *COL1A1* was also highly expressed in TN2, TN3, and TME3 and in the depth of the tumor of ID08 and ID15. Studies suggest that the COL1A1 mediates tumor progression trough mechanism involving EMT, transforming growth factor (TGF)-β, extracellular signal-regulated kinase (ERK), and PI3K/AKT signaling pathways, although further research is needed to unravel the mechanisms by which COL1A1 facilitates cancer cell invasion and proliferation [27]. Regulating *COL1A1* expression directly through miR-133a-3p in oral squamous cell carcinoma suppressed mitosis, proliferation, and invasion of cancer cells [28]. Thirdly, *ASPN*, coding for asporin, is an extracellular matrix proteoglycan involved in cell development and cellular signaling but also, in the scope of cancer, in resistance to growth inhibitors, inhibition of apoptosis, and promotion of cancer metastasis. Indeed, numerous signaling pathways, including TGF-β, Wnt/β-catenin, notch, hedgehog, EGFR, and HER2 have been found to be regulated by asporin. Zhan et al. demonstrated that ASPN is co-localized with HER2 leading to its phosphorylation (p-HER2) [37], which, by regulating the EMT phenotype using the MAPK pathway, promotes thyroid tumor metastasis. In line with our previous hypothesis that activation of CAFs occurs after exposure to anti-HER therapy in some patients [11,13], Itoh et al. showed that the growth of pro-tumorigenic fibroblast from normal fibroblast cells is facilitated by asporin [38]. Finally, studies have shown that overexpression of MUC4, a membrane-bound mucin, induces neoplastic transformation of fibroblasts. Indeed, MUC4 activates Src/Focal adhesion kinase (FAK) and stabilization of HER2 and, thereby, promotes cancer cells survival, invasion, and metastasis. In addition, MUC4 upregulates N-Cadherin expression, which promotes EMT in pancreatic cancer cells [39]. Macha et al. [39] have shown that MUC4 knockdown induces senescence programming pathways, which prevents cell proliferation in vitro and in vivo.

Our study has limitations. The first is the small number of patients included in these analyses. Moreover, EMT can also develop during the evolution of untreated tumors. However, our previous investigations [11] of paired tumor biopsies from control patients had shown no development of EMT during the two-week pre-operative period. Also, the quality of RNA from FFPE sections limits the number of analyzed genes compared with the whole transcriptome data available in case of fresh-frozen samples. Indeed, it is not possible in our analyses to verify whether the absence of a gene’s expression in a spot is linked to a biological or technical condition. Finally, the resolution of the spots (55 µm) in the Visium slides does not allow the study of gene expression at a single-cell level, making the analysis of heterogeneous and mixed tumors undergoing advanced EMT with oligoclusters of tumor cells invading the stromal compartment, as in patient ID08 and ID15, complicated. Indeed, clustering in these tumors makes it impossible to study the stromal and tumor compartments separately with the resolution currently available.

## 4. Material and Methods

### 4.1. Patients

Patients included in this translational research came from the EORTC-90111-24111 trial (NCT01538381). Recruitment, eligibility criteria and randomization were already described [10,11]. Samples for spatial transcriptomic were chosen based on their RNA quality and their cluster classification after afatinib (developing EMT or not), as previously described [11]. The study was conducted in accordance with the International Conference on Harmonization Good Clinical Practice standards and the Declaration of Helsinki. Patients from the CHIRON study (NCT00714649) [12] were also considered as potentially eligible in this translational study.

### 4.2. Tissue Samples

Tumor samples were conserved after curative surgery in the Biobank of the Cliniques universitaires Saint-Luc, Brussels. Paraffin-embedded surgical resection blocks were cut into 5 µm-thick sections and then stained with haematoxylin and eosin (H&E) to firstly confirm the presence of viable invasive tumor cells by a dedicated pathologist, and secondly to select the blocks to be used for spatial transcriptomics analyses. The regions of interest investigated include the surface epithelium, invasive tumor cells and associated stroma, and stop at the tumor cell invasion front.

### 4.3. Assessment of RNA Quantity and Quality

RNA concentration was measured using a Qubit^®^ RNA HS Assay Kit on a Qubit^®^ 4 Fluorometer (Thermo Fisher Scientific Inc., Waltham, MA, USA). RNA quality was evaluated using Agilent High Sensitivity RNA ScreenTape on a 4150 TapeStation instrument (Agilent Technologies, Santa Clara, CA, USA). The percentage of fragments larger than 200 nucleotides (DV200) was evaluated on the basis of the electropherograms using TapeStation Analysis software 5.1.

### 4.4. Spatial Transcriptomics on Surgical Specimens

Visium spatial transcriptomics assays were performed by the lab of Prof. Thierry Voet (KU Leuven, Leuven, Belgium) of the KU Leuven Institute for Single Cell Omics (LISCO). Tissue preparation and sectioning was performed according to 10X Genomics recommendations (Visium Spatial Protocols FFPE—Tissue Preparation Guide, CG000408, RevD). Prior to sectioning, the FFPE blocks were incubated in an ice bath for 10–30 min and 5 µm-thick sections were cut using a microtome. Resulting sections were floated for 40 s in a 42 °C water bath until flat and immediately placed onto the Visium slides. Slides containing sections were dried for 3 h at 42 °C and placed in a desiccator overnight at room temperature to ensure proper drying. They were further stored for a maximum of 1 week before use. Deparaffinization, staining, imaging, decrosslinking, and construction of sequencing libraries were performed according to the manufacturer’s instructions (Visium Spatial Gene Expression for FFPE—deparaffinization, H&E staining, imaging, and decrosslinking, CG000409, RevC; Visium Spatial Gene Expression Reagent Kit for FFPE—User Guide, CG000407, RevD) using the Visium Spatial for FFPE Gene Expression Kit, Human Transcriptome (10X Genomics, 1000338). Briefly, slides with tissue sections were incubated at 60 °C for 2 h and allowed to cool down to room temperature, before deparaffinizing by immersing in xylene for 10 min twice, a 100–70% ethanol series for 3 min each, and finally water for 20 s. Sections were hematoxylin and eosin (H&E) stained, followed by adding 85% glycerol and a coverslip to the slides and imaging on a Nikon NiE microscope with 8-stage at 10X magnification. After coverslip removal, sections were decrosslinked by incubating with 0.1 N HCl for 1 min at room temperature three times, with subsequent washing with TE Buffer (pH 9.0) and incubation with TE Buffer (pH 9.0) at 70 °C for 1 h. Immediately after, overnight Visium probe hybridization was performed, followed by probe ligation, release, and extension the next day. All on-slide reactions were performed in a thermocycler (Bio-rad, C1000 Touch) with a metal slide adapter plate (10X Genomics). Following probe elution, samples were transferred to tubes for amplification, clean-up, and library preparation. Library quality was assessed using an Agilent Technologies Bioanalyzer High Sensitivity kit (Agilent Technologies, 5067-4626). Visium libraries were sequenced on an Illumina NextSeq2000. The sequencing depth was determined by the amount of Visium spots covered by tissue (25,000 reads per spot). Spots corresponding to tumor area were manually selected in the Loupe-Browser (v6.4.1) [40]. SpaceRanger (v2.0.1) was used to process 10X data. Reads were aligned to the human reference sequence GRCh38. Seurat package (v4.3.0) [41] was used to analyze the spatial transcriptomic data. Counts were normalized with the SCTransform function of Seurat package and integrated into a single Seurat object.

Epithelial and stromal score using Puram et al. [42] gene lists were calculated with the function AddModuleScore from the Seurat R package. Differential gene expression analysis was conducted using DESeq2 v1.30 Bioconductor R package. A gene with an adjusted *p*-value (noted as *p* in the text) below 0.05 was defined as differentially expressed. Over-representation enrichment analysis (ORA) was performed with the hallmark gene sets from the Molecular Signatures Database (MSigDB) [43] as hallmark gene sets are known to avoid noise and redundancy in enrichment analyses [44]. Enrichment analysis of the genes commonly upregulated in differential expression analyses was performed using the webtool g:Profiler (https://biit.cs.ut.ee/gprofiler/gost, accessed on 27 September 2023) [45] with the Gene Ontology Molecular Function (GO:MF) and the Reactome from MsigDB. The spatial transcriptomic raw data generated in this study are publicly available in the Gene Expression Omnibus at GSE289908.

## 5. Conclusions

Our study shows for the first time spatial transcriptomic analysis of three surgical specimens of primary HNSCC patients treated pre-operatively with two weeks of afatinib.

The results show tumors expressing EMT-related genes, including one tumor with EMT at an early stage. PDGF and MET pathways, *SNAI2*, *POSTN*, *COL1A1*, *ASPN*, *MUC4*, and others may be targets of choice to prevent the evolution of early stage of EMT in HNSCC tumors treated with anti-EGFR therapy.

## Figures and Tables

**Figure 1 ijms-26-01830-f001:**
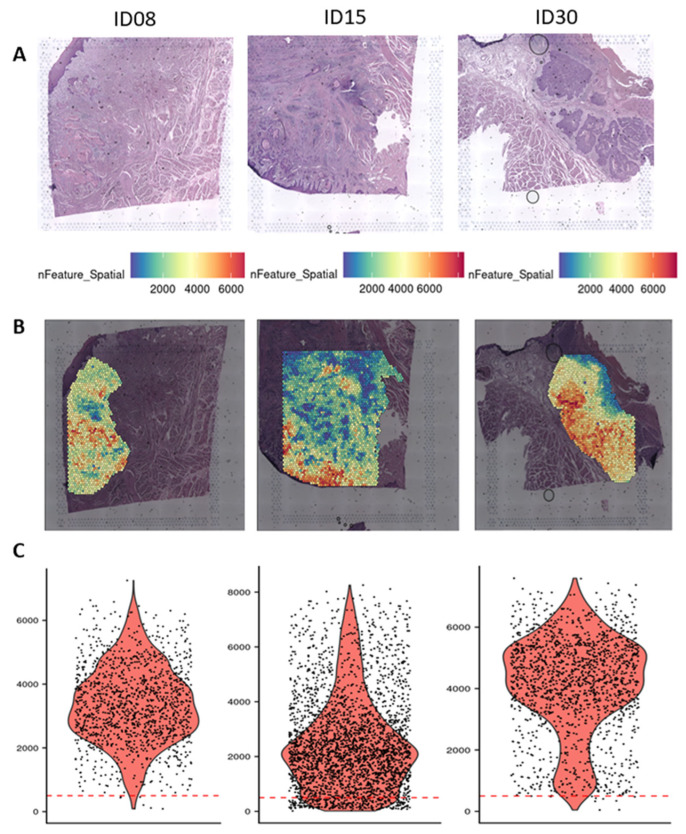
Surgical specimens and nFeature count. (**A**) Surgical specimens of three HNSCC tumors treated pre-operatively during two weeks by afatinib. (**B**) nFeature (i.e., number of gene) count per spots represented on the hematoxylin and eosin slides in a region of interest. A high number of genes is represented in red, a lower number of genes is represented in blue. (**C**) Violin plots showing number of genes per spot in the surgical specimens. The dotted red line represents the 500 arbitrary thresholds.

**Figure 2 ijms-26-01830-f002:**
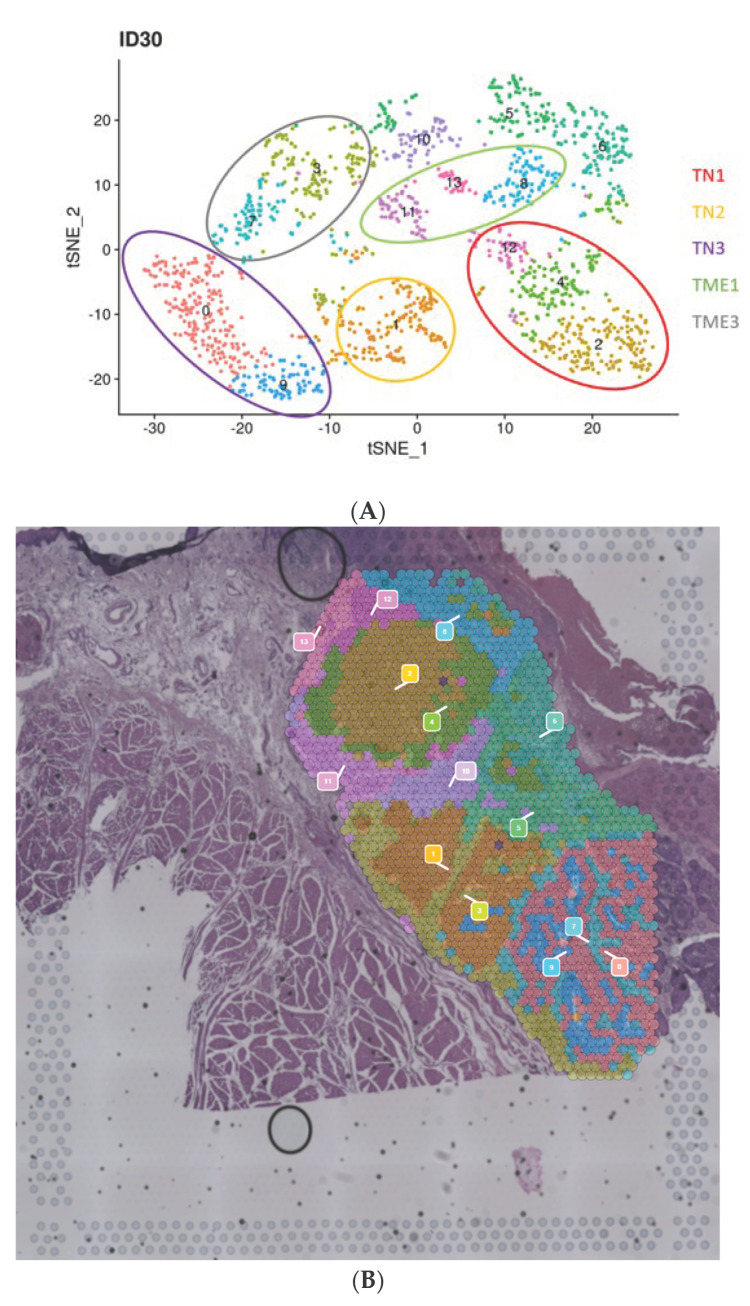
Dimension reduction and clustering for patient ID30. (**A**) t-SNE (t-distributed stochastic neighbor embedding) dimension reduction of spots included in the region of interest of sample ID30. Fourteen clusters were discovered and are represented. Circles represent merge cluster into the subgroups tumor nodules (TN) and the tumor microenvironment (TME). (**B**) Clusters spatially represented on the hematoxylin and eosin of patient ID30. (**C**,**D**) Boxplots (clusters on *x*-axis, scores on *y*-axis) of the epithelial and fibroblasts scores, respectively.

**Figure 3 ijms-26-01830-f003:**
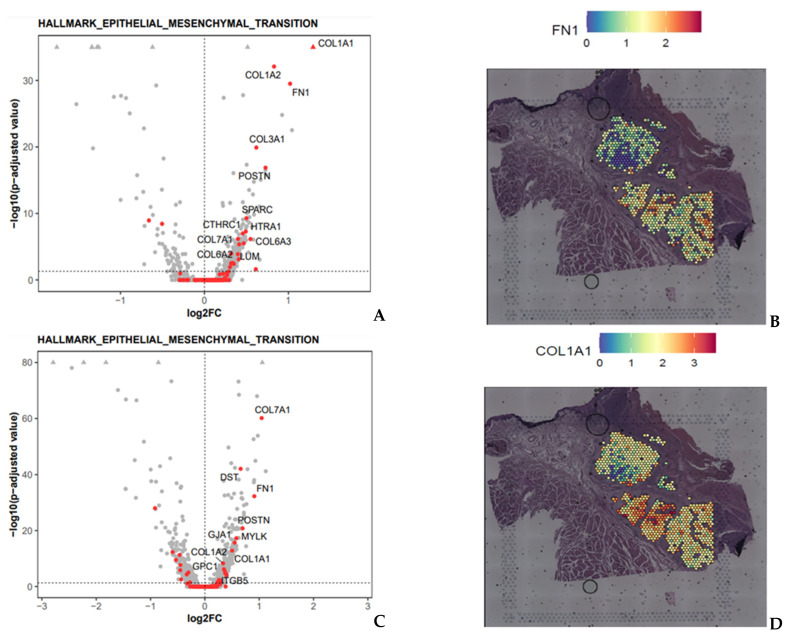
Differential expression analysis of the tumor nodules (TN). (**A**) Volcano plot of the differential expression analysis for TN2 vs. TN1 (*x*-axis: log2 fold-change; *y*-axis: −log10 (adjusted *p*-value). Upregulated genes included in the hallmark epithelial transition gene list are highlighted in red. Genes with adjusted *p*-values < 10–35 are represented by triangles. (**B**) Gene expression of FN1, one of the most upregulated gene in differential expression analysis of TN2 vs. TN1 and TN3 vs. TN1. (**C**) Volcano plot of the differential expression analysis for TN3 vs. TN1 (*x*-axis: log2 fold-change; *y*-axis: −log10 (adjusted *p*-value). Upregulated genes included in the hallmark epithelial transition gene list are highlighted in red. Genes with adjusted *p*-values < 10–80 are represented by triangles. (**D**) Gene expression of COL1A1, one of the most upregulated gene in differential expression analysis of TN2 vs. TN1 and TN3 vs. TN1.

**Figure 4 ijms-26-01830-f004:**
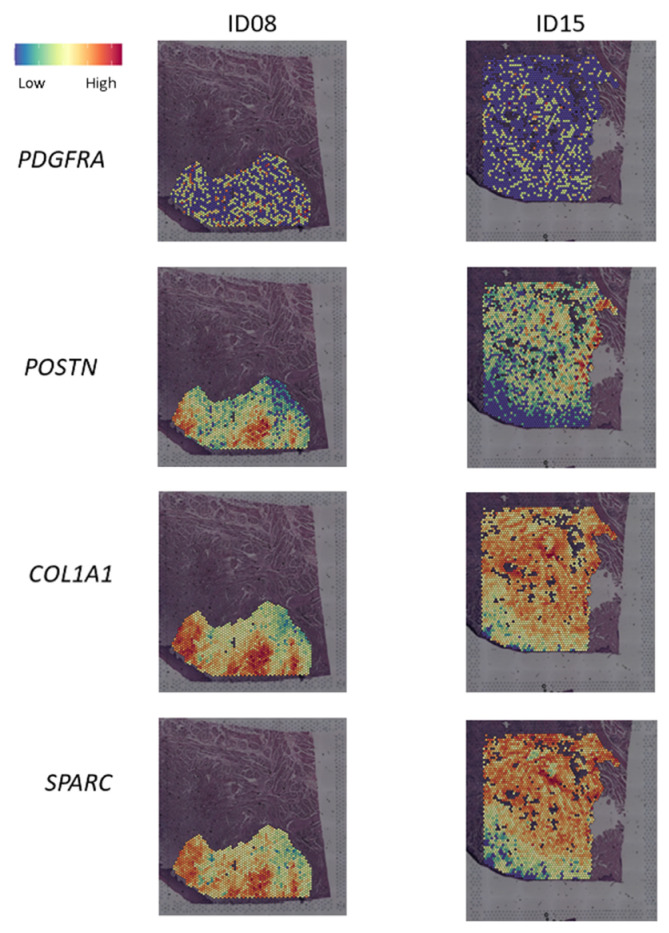
Spatial visualization of genes expression in ID08 and ID15. Some upregulated genes discovered by differential expression analysis in ID30 and included in the 13 common genes between differential expressions analyses of ID30 tumor nodules and afatinib-treated tumor biopsies.

**Figure 5 ijms-26-01830-f005:**
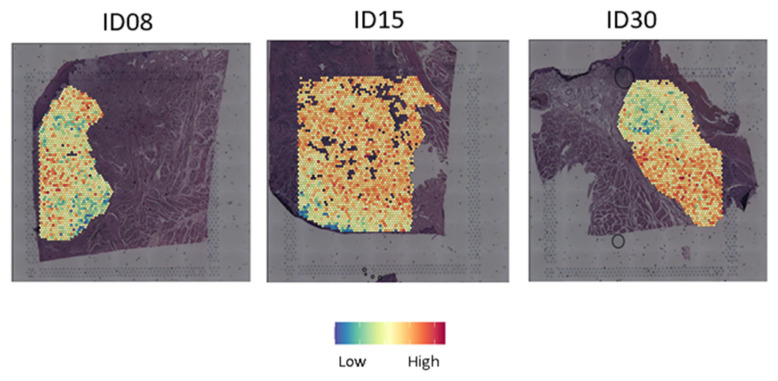
Spatial visualization of expression of 13 common genes between differential expressions analyses of ID30 tumor nodules and afatinib-treated tumor biopsies.

**Figure 6 ijms-26-01830-f006:**
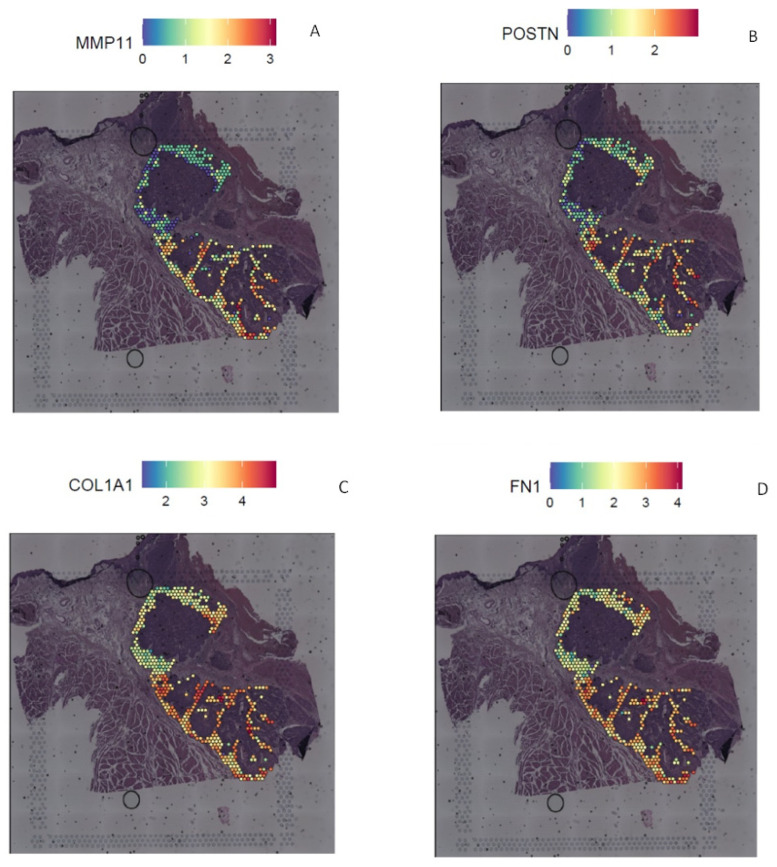
Differential expression analysis of the tumor microenvironment. Representation of the expression of MMP11 (**A**), POSTN (**B**), COL1A1 (**C**) and FN1 (**D**), upregulated genes in the TME3 vs. TME1 differential expression analysis.

**Table 1 ijms-26-01830-t001:** Patient characteristics.

	ID08	ID15	ID30
**Sex**	Male	Female	Male
**Smoker**	Yes	No	Yes
**Pathological status**			
**pT**	T2	T2	T2
**pN**	N0	N2c	N1
**Grade**	Well-differentiated	Moderately differentiated	Poorly differentiated
**Localization**	Oral cavity	Oral cavity	Oropharynx
**RECIST v1.1 after afatinib**	Stable disease	Stable disease	Stable disease
**Relapse**	No	No	Local: 28 months Regional: 38 months

RECIST: Response Evaluation Criteria in Solid Tumors.

## Data Availability

The spatial transcriptomics raw data generated in this study are publicly available in the Gene Expression Omnibus at *** (available after approval of the article).

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
