# Peer review of "Spatial Transcriptomic Analysis of Surgical Resection Specimens of Primary Head and Neck Squamous Cell Carcinoma Treated with Afatinib in a Window-of-Opportunity Study (EORTC90111-24111)"

_ijms, 2025, doi:10.3390/ijms26051830_

Round 1

Reviewer 1 Report

Comments and Suggestions for Authors

Bayert et al.'s study investigates afatinib-induced changes in tumors and the tumor microenvironment in head and neck squamous cell carcinoma. They employ spatial transcriptomics on surgical specimens and RNA sequencing on tumor biopsies from patients enrolled in the EORTC-90111-24111 "window of opportunity" study. Surgical specimens from three patients—two from the epithelial-mesenchymal (EMT) cluster and one from the non-EMT cluster—were analyzed using spatial transcriptomics. Their analysis identified 13 genes involved in cancer and EMT pathways, which were highly expressed in the EMT cluster samples. According to the authors, these genes may represent potential targets to prevent tumor evolution following anti-HER therapy. The study is well-designed, thoroughly described, and of significant scientific interest. Some limitations, such as the small patient sample size, are acknowledged by the authors in their conclusion.

For these reasons, I recommend accepting the article in its current form.

Author Response

The responses are in the PDF.
The manuscript with the changes highlighted in red have been uploaded to the “documents not for publication” section.

Reviewer 2 Report

Comments and Suggestions for Authors

This study evaluated tumor and microenvironment modifications in head and neck squamous cell carcinoma (HNSCC) following afatinib treatment using spatial transcriptomics and RNA sequencing. Conducted as part of the EORTC-90111-24111 window-of-opportunity trial, it aimed to explore tumor evolution and composition under anti-HER therapy. Analysis focused on epithelial-to-mesenchymal transition (EMT) clusters in selected patient samples. Spatial transcriptomics revealed enrichment of EMT-related genes, particularly those involved in PDGF and MET signaling pathways. A set of 13 genes implicated in cancer pathways and EMT was identified, showing high expression in treated tumors. These findings suggest potential therapeutic targets to prevent tumor progression following anti-HER therapy.

There are several issues that need to be resolved.

1. Is there clinical evidence demonstrating that afatinib outperforms cetuximab in terms of adverse effects and patient outcomes?

2. Given that cetuximab remains a first-line treatment for HNSCC, wouldn't it be valuable to investigate the molecular differences between cetuximab-treated and afatinib-treated groups?

3. The analysis results about the tumor microenvironment are limited.

4. The findings lack experimental validation.

Author Response

(The authors gave the same response as above.)

Reviewer 3 Report

Comments and Suggestions for Authors

The study by Beyaert et al entitled “Spatial transcriptomic analysis of surgical resection specimens of primary head and neck squamous cell carcinomas treated with afatinib in a window-of-opportunity study (EORTC90111-24111)” seeks to assess the heterogeneity in response to afatinib treatment in an effort to shed light on potential mechanisms of treatment resistance.  The authors highlight gene expression changes in EMT programs and PDGF pathways following treatment.  Their study digs into 3 of 25 afatinib treated samples (only 13 had suitable RNA quality for analyses); 2 from the cohort with EMT gene expression changes and 1 from the cohort without based on their previous bulk RNA seq analyses (Beyaert et al Clin Can Res 2023).  However, it is unclear why only these three samples were selected for spatial analyses and why none of the control tumour samples were analyzed as this would have been an appropriate comparison to make in order to draw true afatinib treatment-induced gene expression changes.  The novelty of their study is significantly reduced due to much of this spatial transcriptomic work being done as part of their Clinical Cancer Research paper and without the inclusion of appropriate control analyses it is difficult to make any meaningful conclusions.  Suggestions for improvement can be found below.

Major comments:

1.     It would have been more informative to compare equal numbers from both the EMT cluster and non-EMT cluster and perhaps all 13, which in itself is a limited sample size.  What about the control cohort of 5 patients?  Wouldn’t this have been a worthwhile comparator?

2.     In 2.2 of Materials and Methods, the manuscript should indicate how the presence of viable tumor cells is confirmed and how the specific block is selected for spatial transcriptomics. If a pathologist is involved, their role should be explicitly stated.

3.     In Table 1 Patient characteristics the caption includes PET (positron emission tomography), but there is no corresponding information in the table. In addition, relapse status along with progression free survival of the patients is recommended.

4.     In Figures 3A and 3C, the volcano plots should be reformatted to improve data visualization (perhaps could shorten y-axis). As well, gene names for the upregulated genes included in the hallmark epithelial transition gene list should be highlighted in red, not just marked with red dots.

5.     For the 13 common genes identified in bulk RNA-seq analysis (post- vs. pre-afatinib tumor biopsies) and tumor nodule differential expression analyses, a figure, such as a heatmap, should be included to visualize this data. This would provide a better point of reference.

6.     Figure 5 the caption includes ‘some’ upregulated genes discovered by differential expression analysis in ID30. The manuscript should specify why these genes were selected, as this is not explained elsewhere.

7.     The order of text and figures needs to be corrected. Figure 4 appears after Figures 5 and 6.

8.     The authors spend a fair amount of time characterizing the heterogeneity of ID30 (non-EMT cluster patient).  Do any of the findings related to the heterogeneity in EMT signatures correlate with p-EGFR levels?  Is this truly due to differences in responsiveness of the nodules or are other microenvironmental factors at play?

9.     Conclusion:  “our study shows for the first time spatial transcriptomic data from three surgical specimens…”  is not true as they had previously shown this in their Clinical Cancer Research paper Beyaert SP et al Clin Can Res 2023 and needs to be corrected.

Minor Comments

1.     Puram et al.'s gene list is frequently referenced and should be included in the supplementary materials, as it is used for epithelial, stromal, and fibroblast score for clustering of spots.

2.     For Figure 1B, the color scale is difficult to interpret. Provide units or a clear indication i.e., the scale ranges from 0 to 2, with 2 represents the highest number of genes per spot.

3.     In 3.2 Quality Control, the authors should provide the DV200 value for the RNA of each of the three samples, not only the mean DV200 for the analysis.

4.     The manuscript does not mention how the region of interest was identified in the slides of the samples. This information should be included in the Materials and Methods section.

5.     In section 3.4, for the TN2 vs. TN1 & TN3 vs. TN1 differential expression analyses, only the common upregulated genes were included. Consider also addressing the downregulated genes.

6.     In Figure 6, correct the spelling of ‘tumor’ in the caption.

Author Response

(The authors gave the same response as above.)

Round 2

Reviewer 2 Report

Comments and Suggestions for Authors

The concerns raised have not been adequately addressed.

Reviewer 3 Report

Comments and Suggestions for Authors

The authors have addressed my concerns.